# Experimental evidence for delayed post-conflict management behaviour in wild dwarf mongooses

**Amy Morris-Drake[1]\*, Julie M Kern[2], Andrew N Radford[1]**

[1]School of Biological Sciences, University of Bristol, Bristol, United Kingdom; [2]School of Environmental and Rural Science, University of New England, Armidale, Australia

**Abstract** In many species, within-group conflict leads to immediate avoidance of potential aggressors or increases in affiliation, but no studies have investigated delayed post-conflict management behaviour. Here, we experimentally test that possibility using a wild but habituated population of dwarf mongooses (*Helogale parvula*). First, we used natural and playback-simulated foraging displacements to demonstrate that bystanders take notice of the vocalisations produced during such within-group conflict events but that they do not engage in any immediate post-conflict affiliative behaviour with the protagonists or other bystanders. We then used another playback experiment to assess delayed effects of within-group conflict on grooming interactions: we examined affiliative behaviour at the evening sleeping burrow, 30–60 min after the most recent simulated foraging displacement. Overall, fewer individuals groomed on evenings following an afternoon of simulated conflict, but those that did groomed more than on control evenings. Subordinate bystanders groomed with the simulated aggressor significantly less, and groomed more with one another, on conflict compared to control evenings. Our study provides experimental evidence that dwarf mongooses acoustically obtain information about within-group contests (including protagonist identity), retain that information, and use it to inform conflict-management decisions with a temporal delay.

**\*For correspondence:**
am9162@bristol.ac.uk

**Competing interest:** The authors declare that no competing interests exist.

## Introduction

Conflicts of interest are common in social species, with disagreements between group members arising over access to mates or food, synchronisation of group activities, and the direction of travel (*Aureli et al., 2002*; *Conradt and Roper, 2009*; *Hardy and Briffa, 2013*). Within-group conflict, especially if it escalates to aggression, can be costly in terms of injury and mortality, time and energy expenditure, increased stress, and disrupted social relationships (*Aureli, 1997*; *Aureli et al., 2002*; *de Waal, 2000*). Conflict-management strategies that minimise these costs, either by reducing the likelihood of aggressive escalation in the first place or by mitigating the consequences of such physical contests when they do arise, have therefore evolved in many species (*Aureli et al., 2002*; *Aureli and de Waal, 2000*). Much of the early work on post-conflict behaviour focussed on interactions between the protagonists (the aggressor and the target): many studies have documented increases in affiliation between former opponents in the aftermath of a contest (reconciliation; *Aureli et al., 2002*; *de Waal, 2000*; *de Waal and van Roosmalen, 1979*), although there are also examples of victims avoiding aggressors (wariness; *Benkada et al., 2020*; *Kutsukake and Clutton-Brock, 2008*; *Sommer et al., 2002*). More recently, attention has shifted to the involvement of bystanders (contest nonparticipants) in post-conflict behaviour. Considering bystanders highlights the potentially group-wide effects of dyadic within-group conflicts and a wider range of post-conflict management strategies than would be apparent from a focus on just the protagonists (*De Marco et al., 2010*; *Schino and Sciarretta, 2015*), thus providing additional insights into the dynamics of social relationships between

**eLife digest** Social animals that live in groups often have disagreements over access to mates and food. Even fleeting in-group disputes can be costly, disrupting relationships, wasting time and energy, or causing injury if aggression escalates. So, much like humans, many social animals, including primates, birds and dogs, have evolved conflict management strategies to prevent and resolve in-group disagreements. In the immediate aftermath of a conflict, this usually involves changes in the interactions between those involved in the disagreement, or between bystander groupmates and either the victim or aggressor.

Less is known about whether social animals can recall past disputes and if they can use conflict management strategies some time after a quarrel has occurred. That is, do aggressive interactions between groupmates influence later social decisions of bystanders in the group?

To investigate, Morris-Drake et al. studied groups of wild dwarf mongooses (*Helogale parvula*) that have become accustomed to living alongside humans in Limpopo Province, South Africa. Dwarf mongooses live in groups of up to 30 individuals, with one dominant breeding pair and lower-ranked helpers. When disagreements arise over food, an aggressor growls deeply and hip-slams the victim away from their foraging patch, stealing the victim's prey in the process. Victims often produce high-pitched squeals in retreat.

Using recordings of these calls, Morris-Drake et al. devised a field experiment to investigate how mongooses responded to nearby conflicts between other group members. Recordings simulating a conflict over food were played to groups of foraging mongooses over the course of an afternoon, so that group members effectively heard what sounded like repeated squabbles between two out-of-sight individuals. Similar to natural conflicts, the mongooses did not engage in any obvious conflict management behaviour immediately after hearing these disputes. But when the group returned to their sleeping burrow that evening, subordinate group members shunned the perceived aggressors from grooming, a key social activity.

This work provides evidence that dwarf mongooses keep tabs on conflicts that occur between groupmates. It shows these animals can extract information about conflicts from vocal cues alone and that bystanders use this information when making later social decisions impacting group dynamics. It also adds to growing evidence from baboons, monkeys and chimpanzees that social animals can remember past events and take these into account when interacting with groupmates.

---

groupmates (*Aureli and de Waal, 2000*). Multiple studies have now documented bystander-initiated affiliation with the victim as a means of avoiding redirected aggression (self-protection) or of providing substitute reconciliation or consolation (*Fraser et al., 2009*; *Fraser et al., 2008*; *Schino and Marini, 2012*; *Wittig and Boesch, 2010*). There is also some evidence of bystander-initiated affiliation with the aggressor, which could function as appeasement to reduce the likelihood of redirected aggression (*Cordoni and Palagi, 2015*; *Palagi et al., 2008*; *Pallante et al., 2018*), and group-wide post-conflict affiliation among bystanders, perhaps to reduce conflict-induced stress (*De Marco et al., 2010*; *Judge and Mullen, 2005*). However, to the best of our knowledge, this research has focussed solely on interactions that occur in the immediate aftermath (usually within 10 min) of an aggressive within-group contest; the possibility of delayed post-conflict management behaviour has not been explored.

There is increasing experimental evidence that nonhuman animals can remember past events and use information from them when making social decisions later (*Carter and Wilkinson, 2013*; *Kern and Radford, 2018*; *Seyfarth and Cheney, 1984*; *Wittig et al., 2014*). This includes conflict-management decisions about whether to get involved in an aggressive interaction. For example, baboons (*Papio hamadryas ursinus*) were more likely to offer support in aggressive interactions to individuals they had groomed with earlier (mean: 22 min before; range 10–55 min), evidenced by a move towards playbacks of grunt calls given during conflicts (*Cheney et al., 2010*). Similarly, vervet monkeys (*Chlorocebus pygerythrus*) were more likely to offer coalitionary support to a groupmate in a conflict if they had groomed together within the last hour (*Borgeaud and Bshary, 2015*). Other studies have shown that individuals can use knowledge of previous agonistic interactions to inform how best to respond in subsequent aggressive encounters. For instance, chimpanzees (*Pan troglodytes*) that had been involved in an unreconciled conflict earlier in the day (ca 2 hr before) reacted aversively to the

playback of an aggressive bark from their former opponent's bond partner (a third-party individual likely to offer aggressive support to the former opponent; *Wittig et al., 2014*). Moreover, it was recently shown that bystander wasps (*Polistes fuscatus*) were more aggressive towards individuals that they had observed to be less aggressive in a previous (10–30 min earlier) fight with a third party (*Tibbetts et al., 2020*). It is thus plausible that post-conflict decisions about the avoidance of protagonists and affiliation with groupmates could also occur some time after the relevant within-group contests. Investigating the capacity for delayed management behaviour is important because it is thought to be cognitively challenging to use social information gathered in the past, especially where there is reliance on memories of the actions of particular individuals (*Frith and Frith, 2012*), as other sources of personal and third-party information would likely arise in the interim (*Wittig et al., 2014*).

To make behavioural decisions, animals obtain information about social interactions using a variety of sensory modalities. Most research considering social monitoring of within-group conflicts has focussed on situations where individuals have seen the interaction; hence, bystanders are commonly defined as individuals who have observed the encounter (*Schino and Sciarretta, 2015*). But for those species living in visually occluded environments, those where group members can be scattered over large distances or those that forage in a way that prevents simultaneous vigilance, acoustic cues can be a valuable source of social information (*Bradbury and Vehrencamp, 2011*). Numerous species vocalise during or at the end of within-group contests (*Bertram et al., 2010*; *Slocombe et al., 2010*). For example, chimpanzees and rhesus monkeys (*Macaca mulatta*) produce screams when experiencing aggression (*Gouzoules et al., 1984*; *Slocombe et al., 2010*), whilst little blue penguins (*Eudyptula minor*) give specific calls after a contest is finished (*Waas, 1990*). These vocalisations likely provide bystanders with valuable information about the occurrence of within-group conflicts as well as about the group members that have potentially been involved and the outcome (*Gouzoules et al., 1984*; *Slocombe and Zuberbühler, 2007*; *Szipl et al., 2017*; *Whitehouse and Meunier, 2020*). Moreover, they can be used in playbacks to test post-conflict behaviour experimentally.

Here, we investigate post-conflict management behaviour, including the possibility that it occurs with a delay (ca 30–60 min later), in wild dwarf mongooses (*Helogale parvula*); the study population has been habituated to close human presence, facilitating detailed observations and field-based experiments (*Kern and Radford, 2018*; *Morris-Drake et al., 2019*). Dwarf mongooses live in cooperatively breeding groups of up to 30 individuals, comprising a dominant breeding pair (hereafter 'dominant' individuals) and non-breeding subordinate helpers (hereafter 'subordinate' individuals) of both sexes (*Rasa, 1977*). The most prevalent affiliative behaviour in dwarf mongoose groups is allogrooming (hereafter 'grooming'), which underpins the strength of relationships between group members (*Kern and Radford, 2021*; *Kern and Radford, 2016*), increases following stressful situations such as intergroup interactions (*Morris-Drake et al., 2019*), and is traded as a reward for cooperative behaviour (*Kern and Radford, 2018*). Within-group aggressive interactions take two main forms: relatively rare targeted aggression, which usually acts to reinforce rank and is mainly due to reproductive conflict (*Rasa, 1977*), and relatively common foraging displacements, when a higher-ranking individual displaces a lower-ranking group member from a foraging patch and steals their prey (*Sharpe et al., 2016*; *Sharpe et al., 2013*). Foraging displacements generally involve the following behavioural sequence: the higher-ranking individual produces deep growls as it approaches the lower-ranking group member; the former then hip-slams the latter away from the food resource; and the displaced individual typically produces high-pitched squeals whilst it retreats (*Sharpe et al., 2016*; *Sharpe et al., 2013*). We determined whether vocal cues of within-group conflict elicit immediate or delayed behavioural responses (avoidance or changes in affiliation) by non-participant group members. We focussed on data collection of bystanders because it is not ecologically valid to consider how protagonists respond to their own calls.

## Results

### Immediate behavioural responses of bystanders to within-group conflict

We initially used both observational data and a playback experiment to investigate whether bystanders take notice of conflict between groupmates (evidenced by an increase in vigilance) and if they engage in affiliative interactions (grooming) as post-conflict management behaviour in the

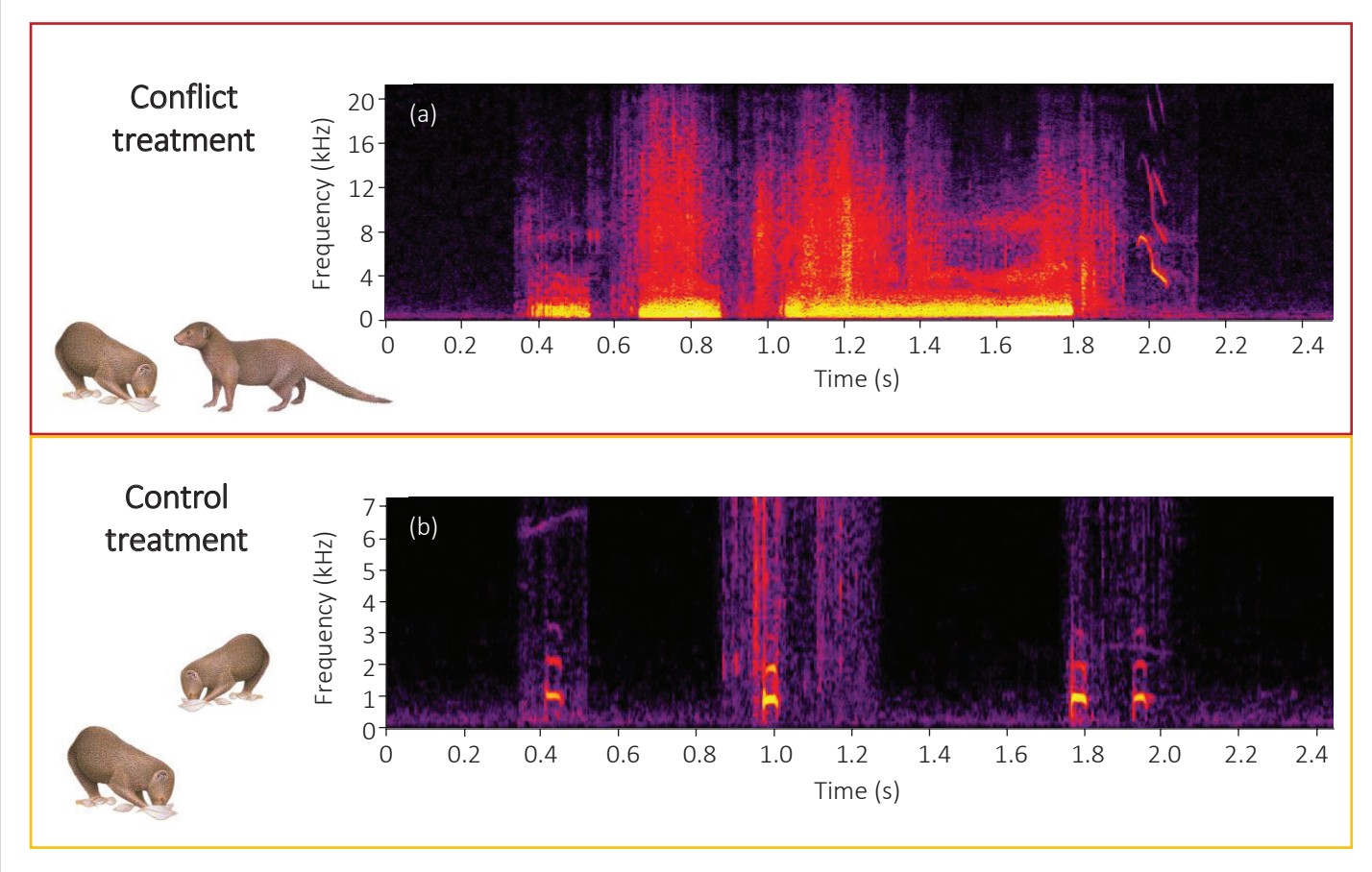

**Figure 1.** Spectrograms of the final sections of example (**a**) conflict and (**b**) control playback tracks. A conflict track concluded with three growls from a dominant aggressor followed by a squeal from a subordinate target, whilst a control track concluded with three close calls from the same dominant individual followed by one close call from the same subordinate individual as in the matched conflict track. Spectrograms were created in Raven Pro 1.5 using a 1024 point fast Fourier Transform (Hamming window, 75 % overlap, 2.70 ms time resolution, 43 Hz frequency resolution).

immediate aftermath (full details in 'Materials and methods'). To collect data relating to natural foraging displacements (which occur at a mean ± SE observer-detected rate of 2.6 ± 0.2 events per 3 -hr observation session, range = 0–10, N = 127 observation sessions across eight groups), we conducted focal watches on foraging subordinates in two situations: immediately after the human observer heard a foraging displacement (conflict situation) and on a matched occasion when there had been no foraging displacement for at least 10 min (control situation). Paired data were collected from 16 subordinates in six groups, with conflict and control focal watches counterbalanced in order between individuals. To test experimentally the immediate responses of bystanders, and to isolate the importance of foraging-displacement vocalisations as a cue to conflict occurrence, we presented 17 foraging subordinates in eight groups with two playback treatments in a matched, counterbalanced design (Experiment 1). The conflict treatment entailed an initial playback of close calls from a dominant individual and a subordinate individual from the same group as the focal individual, followed by a playback of the dominant growling and the subordinate squealing (simulating a foraging displacement); the control treatment entailed the playback of close calls from the same two individuals for the same duration as a full conflict-treatment playback track (*Figure 1*). Foraging dwarf mongooses produce continuous low-amplitude close calls, which likely enable groupmates to stay in contact; there is no evidence that they have an aggressive function (*Kern and Radford, 2013*; *Sharpe et al., 2013*). We chose for the playback the combination of a dominant individual as the aggressor and a subordinate individual as a target because this is the most common dyadic pairing observed in natural foraging displacements (74.3 % of 740 events in 12 groups).

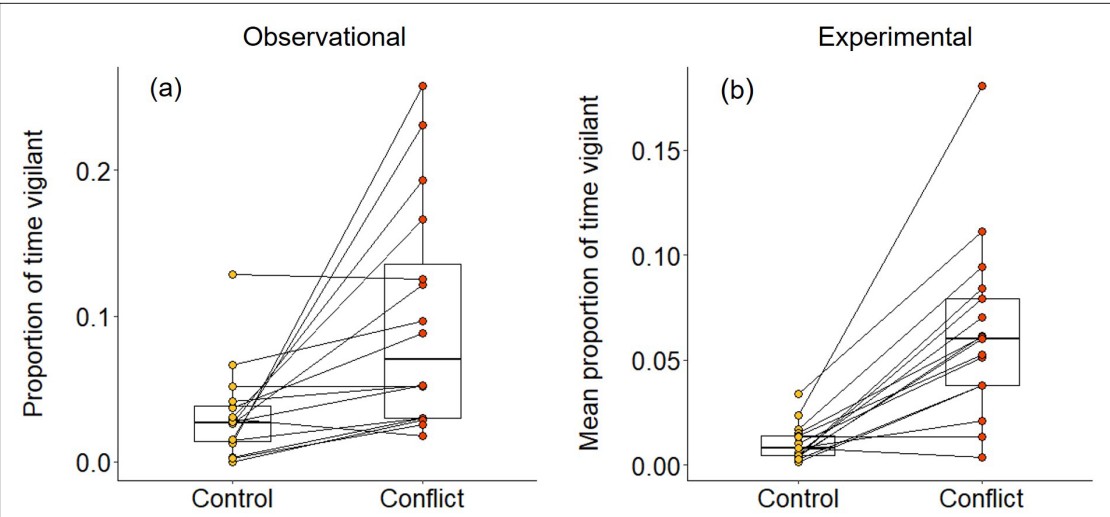

**Figure 2.** Immediate effect of within-group conflict on dwarf mongoose vigilance behaviour. Compared to control situations, (**a**) natural foraging displacements (observational; N = 16 individuals in six groups) and (**b**) simulated foraging displacements (experimental; N = 17 individuals in eight groups) both resulted in a greater proportion of time spent vigilant by foragers in the subsequent 2–3 min. Shown in both panels are boxplots with the median and quartiles; whiskers represent data within quartiles ± 1.5 times the interquartile range. Values for each individual are given as circles, with lines connecting data from the same individual; in some instances, more than one individual has the same value, hence the number of lines can appear less than the stated sample size.

The online version of this article includes the following source data for figure 2:

**Source data 1.** Proportion of time spent vigilant by focal individuals following a natural (N=16) or simulated (N=17) within-group conflict or matched-control situation.

In 2–3 min following both natural foraging displacements (Wilcoxon signed-rank test: Z = 3.154, N = 16, Monte Carlo p<0.001; *Figure 2a*) and those simulated by playbacks (Z = 3.527, N = 17, p<0.001; *Figure 2b*), focal foragers spent a significantly greater proportion of time vigilant than in matched-control, non-conflict situations. The increased vigilance following foraging displacements indicates that bystanders take notice of conflict between groupmates; the experimental results demonstrate that the vocal cues are sufficient to trigger this reaction. However, the focal individual did not engage in any post-conflict grooming in the 5 min following either natural or simulated foraging displacements; grooming is generally rare (ca 10 % of bouts) during foraging periods in dwarf mongooses (*Kern and Radford, 2018*). Thus, dwarf mongoose bystanders do not appear to engage in post-conflict affiliative behaviour in the immediate aftermath of hearing a within-group contest.

## Delayed behavioural responses of bystanders to within-group conflict

To test if there were delayed effects of within-group conflict on affiliative behaviour (grooming), we conducted a second repeated-measures playback experiment on eight groups (Experiment 2, *Figure 3*; full details in 'Materials and methods'). The general experimental design followed *Kern and Radford, 2018*. In each trial session, we either simulated an increase in the conflict between a dominant (aggressor) and a subordinate (target) group member through playback of their foraging-displacement vocalisations (conflict treatment) or played back just the close calls of those individuals for an equivalent period (control treatment). Trials were on separate days with treatment order counterbalanced between groups. In each trial, six to nine playbacks (mean ± SE: 8.5 ± 0.2, N = 16 trials) were carried out over the course of 3 hr in the afternoon whilst the group were foraging and before they moved towards their evening sleeping refuge (mean ± SE period between the final playback and first grooming bout at the sleeping refuge: 37 ± 5 min, N = 16 trials); individual playbacks were as in Experiment 1 with different tracks played each time. At the refuge, we collected data ad libitum on all adult grooming interactions, including the identity of those involved and bout duration; each bout was always between just two individuals and generally mutual (both parties approaching each other and grooming, without an obvious initiator). If within-group conflict does have delayed effects on affiliative behaviour, we expected an increase in the occurrence of foraging displacements to result in changes

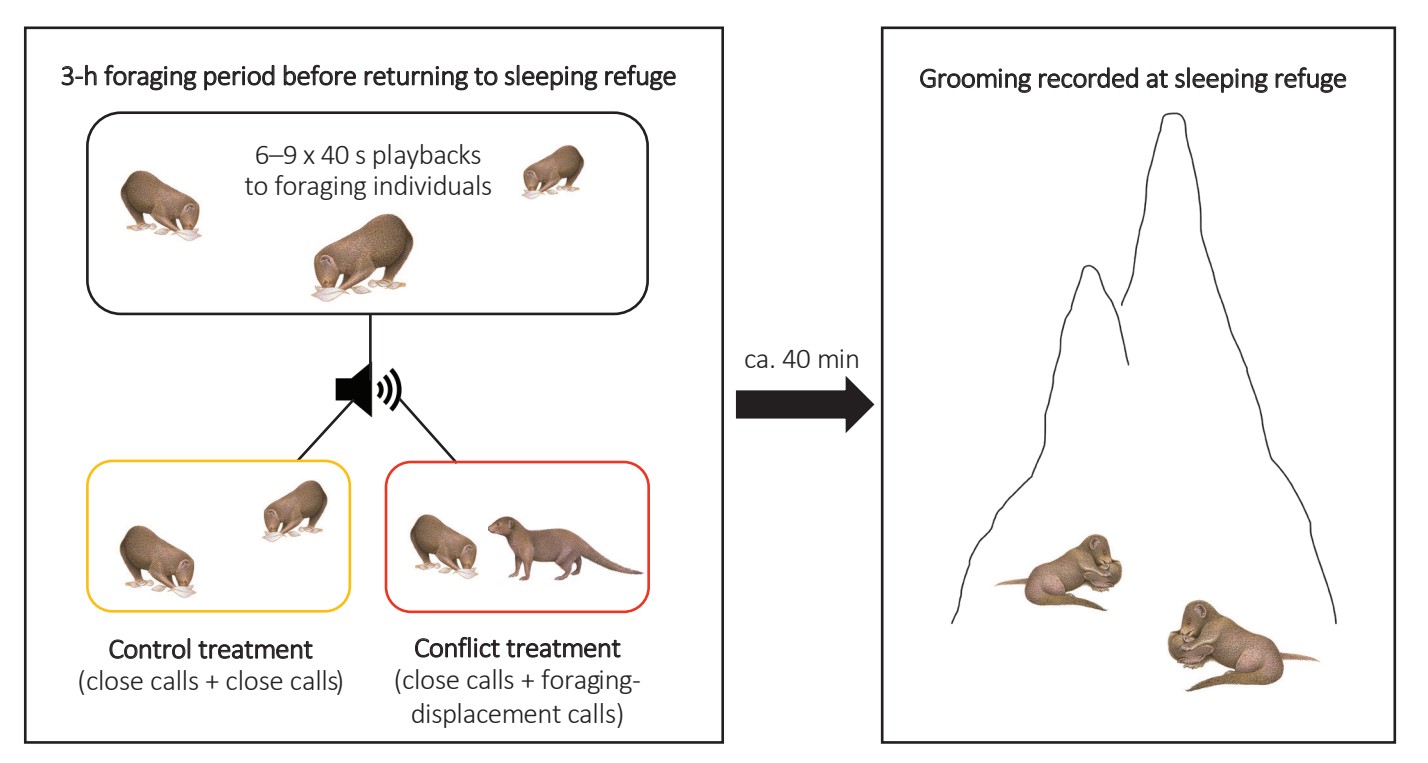

**Figure 3.** Illustration of the protocol for Experiment 2. Within-group conflicts between a dominant aggressor and a subordinate target were simulated during conflict-treatment afternoons using playbacks of foraging-displacement calls, with only close calls of the same individuals played back in control sessions. All grooming at the evening sleeping refuge was subsequently recorded following both treatments.

in evening grooming levels; 90 % of grooming bouts occur at the sleeping refuge (N = 6376 bouts, 174 individuals; *Kern and Radford, 2018*).

Overall, we found that group members were significantly less likely to be involved in grooming interactions in the evenings following conflict afternoons compared to control afternoons (generalised linear mixed model [GLMM]: $\chi^2$ = 5.401, df = 1, p = 0.020; *Table 1a*; *Figure 4a*). However, when considering only those individuals that engaged in grooming, they spent a significantly greater proportion of time doing so on evenings when there had been an earlier simulated increase in conflict compared to control evenings ($\chi^2$ = 15.873, df = 1, p<0.001; *Table 1b*; *Figure 4b*). This was because these individuals were grooming more frequently ($\chi^2$ = 8.010, df = 1, p = 0.005; *Table 1c*) and for longer per bout (linear mixed model [LMM]: $\chi^2$ = 3.958, df = 1, p = 0.047; *Table 1d*) after a simulated increase in conflict compared to control conditions. These results indicate that there is an overall response to simulated conflict within the group, but we also made some specific predictions. Assuming that aggressors and targets can be identified from their vocalisations—which has been demonstrated for dwarf mongoose close calls (*Sharpe et al., 2013*), recruitment calls (*Kern and Radford, 2016*), and surveillance calls (*Kern and Radford, 2018*)—we predicted that subordinates might engage in either less grooming (due to wariness) or more grooming (as possible appeasement) with aggressors, and that they might engage in more grooming with targets (as possible consolation).

We found evidence that simulating aggressive behaviour by a dominant individual during the afternoon resulted in subordinates engaging in less grooming with it at the sleeping refuge that evening. Following conflict trials, subordinates groomed with the dominant pair for a smaller proportion of time than after control trials (Wilcoxon signed-rank test: Z = 2.240, N = 8, p = 0.021). The reduced affiliative engagement by subordinates was driven by a change in behaviour towards the simulated aggressor specifically: subordinates engaged in significantly less grooming with the simulated aggressor on conflict evenings compared to control evenings (proportion of time: Z = 2.521, N = 8, p = 0.008; *Figure 5a*; proportion of subordinates: Z = 2.201, N = 8, p = 0.033; *Figure 5c*), but there was no such treatment difference in the grooming of subordinates with the dominant whose calls were not played

**Table 1.** Output from mixed models investigating the grooming behaviour of adult dwarf mongooses at the evening refuge.

All models contained treatment (conflict, control) as a fixed effect (the reference level in the table is 'conflict'), with Individual ID nested within Group ID as random effects. The first model (generalised linear mixed model (GLMM) with a binomial error distribution and logit-link function) examined (a) whether an individual was involved in a grooming bout (Yes or No). Subsequent models focussed on those individuals that did participate in grooming, examining (b) the proportion of time spent grooming (GLMM with a beta error distribution and logit-link function), (c) the rate of grooming interactions (GLMM with a Poisson error distribution, log-link function, and log(duration) as an offset term to account for differences in the time available for grooming), and (d) the log-transformed mean grooming-bout duration (linear mixed model (LMM) with a Gaussian error distribution). Significant fixed effects shown in bold; variance ± SD reported for random effects is shown in italics.

| | Effects | Estimate ± SE | df | $\chi2$ | p |
|---|---|---|---|---|---|
| **(a) Individual involvement in grooming** | | | | | |
| Random effects | *Group ID* | *0.919 ± 0.959* | | | |
| | *Individual ID in Group* | *<0.001 ± <0.001* | | | |
| Minimal model | (Intercept) | 1.267 ± 0.497 | | | |
| | **Treatment (Conflict)** | **1.106 ± 0.488** | **1** | **5.401** | **0.020** |
| | | | | | |
| **(b) Proportion of time spent grooming** | | | | | |
| Random effects | *Group ID* | *0.187 ± 0.433* | | | |
| | *Individual ID in Group* | *0.065 ± 0.255* | | | |
| Minimal model | (Intercept) | −1.544 ± 0.190 | | | |
| | **Treatment (Conflict)** | **−0.697 ± 0.164** | **1** | **15.873** | **<0.001** |
| | | | | | |
| **(c) Rate of grooming bouts** | | | | | |
| Random effects | *Group ID* | *<0.001 ± 0.018* | | | |
| | *Individual ID in Group* | *0.195 ± 0.442* | | | |
| Minimal model | (Intercept) | −1.317 ± 0.102 | | | |
| | **Treatment (Conflict)** | **−0.296 ± 0.105** | **1** | **8.010** | **0.005** |
| | | | | | |
| **(d) Mean grooming-bout duration** | | | | | |
| Random effects | *Group ID* | *0.047 ± 0.217* | | | |
| | *Individual ID in Group* | *0.067 ± 0.258* | | | |
| Minimal model | (Intercept) | 3.257 ± 0.104 | | | |
| | **Treatment (Conflict)** | **−0.167 ± 0.083** | **1** | **3.958** | **0.047** |

back (proportion of time: Z = 0.105, N = 8, p = 1; *Figure 5b*; proportion of subordinates: Z = 0.813, N = 8, p = 0.499; *Figure 5d*). Moreover, on those occasions where individuals did groom, bout durations were somewhat shorter on conflict evenings compared to control evenings for grooming involving simulated aggressors (mean ± SE duration, post-control: 34 ± 11 s; post-conflict: 23 ± 5 s; N = 4 pairs of trials), while the reverse was true for grooming involving the matched dominant (post-control: 28 ± 8 s; post-conflict: 34 ± 8 s; N = 4 pairs of trials); small sample sizes precluded statistical analysis.

We also found some evidence that increasing within-group conflict during the afternoon resulted in more evening grooming between subordinates. When considering all bouts between subordinate group members, there was no significant treatment difference in the proportion of time spent grooming

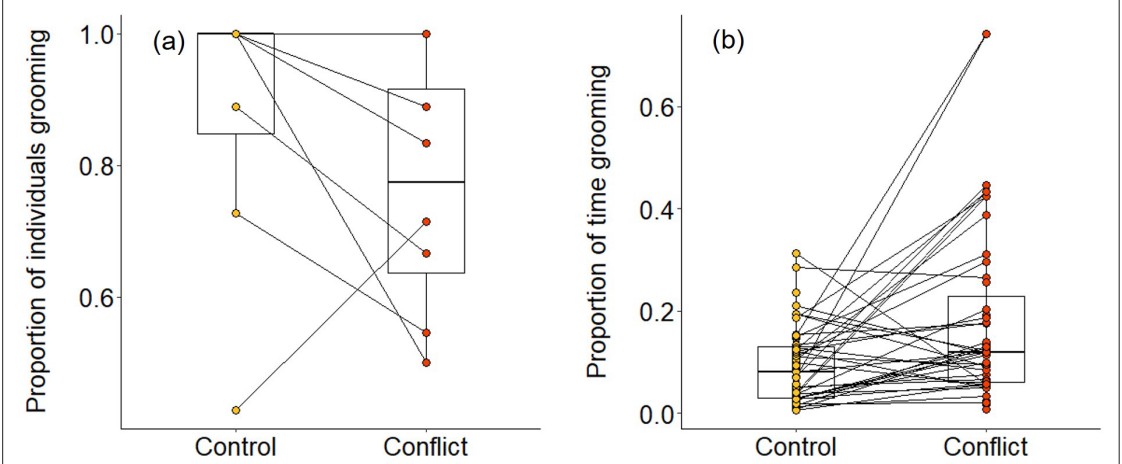

**Figure 4.** Delayed effect of experimentally increased within-group conflict on dwarf mongoose grooming behaviour. Compared to control afternoons, those with simulated additional foraging displacements resulted in (**a**) a smaller proportion of adult group members engaged in evening grooming behaviour (N = 8 groups) but (**b**) a greater proportion of time engaged in grooming by those individuals that did any grooming (N = 63 individuals in eight groups). Shown in both panels are boxplots with the median and quartiles; whiskers represent data within quartiles ± 1.5 times the interquartile range. Values for each group or individual are given as circles, with lines connecting data from the same group or individual; orphan points, where an individual only groomed in one treatment, are also plotted. In some instances, more than one group or individual has the same value, hence the number of lines can appear less than the stated sample size.

The online version of this article includes the following source data for figure 4:

**Source data 1.** Proportion of adult group members engaged in evening grooming behaviour (N=8 groups) and proportion of time engaged in grooming by those individuals that did any grooming (N=63 individuals in eight groups) following afternoons with simulated within-group conflict or matched-control afternoons.

(Wilcoxon signed-rank test: Z = 1.540, N = 8, p = 0.146), but subordinate–subordinate grooming bouts were, on average, significantly longer on conflict evenings compared to control evenings (Z = 2.366, N = 7, p = 0.015; *Figure 6a*). Considering bouts involving particular individuals, there were indications that targets might receive a conflict-driven increase in grooming from other subordinates not seen for preselected control subordinates (those whose squeals had not been played back), but no statistically significant differences. The proportion of time grooming that involved the simulated target was doubled on conflict evenings (mean ± SE: 0.31 ± 0.09) compared to control evenings (0.15 ± 0.06; Z = 1.572, N = 8, p = 0.156; *Figure 6b*), whereas there was, if anything, a decrease for the preselected control subordinate (control: 0.37 ± 0.12; conflict: 0.28 ± 0.09; Z = 0.280, N = 8, p = 0.843; *Figure 6c*). The treatment difference in mean bout duration was also greater for grooming involving simulated targets (36 ± 14 s, N = 3 pairs of trials) than that involving preselected control subordinates (22 ± 24 s, N = 3 pairs of trials), but too few matched evenings involved the relevant individuals to allow statistical testing.

## Discussion

Dwarf mongoose bystanders did not engage in any obvious post-conflict affiliation in the immediate aftermath of natural or simulated foraging displacements involving a dominant and subordinate group member, but did adjust their later grooming behaviour at the evening sleeping refuge following a simulated increase in within-group conflict during the afternoon. The increase in the average duration of later subordinate–subordinate grooming is in line with the increase in bystander–bystander grooming seen in some species in the immediate aftermath of a contest (*Judge and Mullen, 2005*). Such affiliation could reduce the group-wide social anxiety induced by aggression (*De Marco et al., 2010*; *Judge and Bachmann, 2013*; *Schino and Sciarretta, 2015*). The later reduction in grooming of aggressors by bystanders is, to our knowledge, the first evidence for a change in this direction; some previous studies have documented increased grooming of aggressors by bystanders in the immediate aftermath of a single contest (*Cordoni and Palagi, 2015*; *Palagi et al., 2008*; *Pallante et al., 2018*), whilst a few others have found no evidence for such an increase (*Judge, 1991*; *Romero*

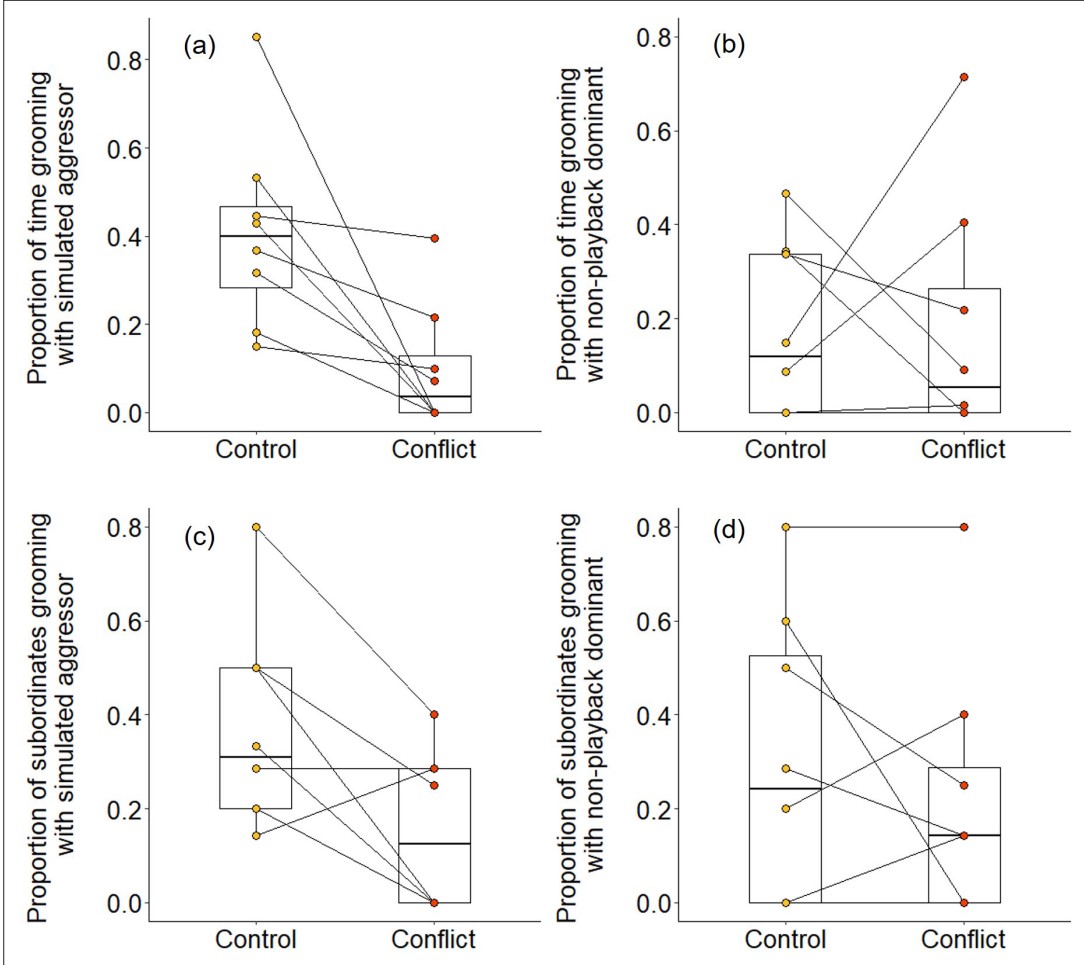

**Figure 5.** Delayed effect of experimentally increased within-group conflict on the grooming between subordinate bystanders and the simulated aggressor. Compared to control afternoons, those with simulated additional foraging displacements between a dominant aggressor and a subordinate target resulted in (**a**) a smaller proportion of time engaged in evening grooming by subordinate bystanders with the dominant aggressor but (**b**) no such treatment difference in the proportion of time that subordinate bystanders groomed with a non-playback dominant. At least in part, this was because (**c**) a smaller proportion of subordinate bystanders groomed with the dominant aggressor in the evening of conflict afternoons compared with control ones, but (**d**) there was no such treatment difference in the proportion of subordinates involved in grooming with a non-playback dominant. Shown in all panels are boxplots with the median and quartiles; whiskers represent data within quartiles ± 1.5 times the interquartile range. Values for each group are plotted separately (N = 8), with lines connecting data from the same group; in some instances, more than one group has the same value, hence the number of lines can appear less than eight.

The online version of this article includes the following source data for figure 5:

**Source data 1.** Proportion of time engaged in evening grooming by subordinate bystanders and proportion of those individuals involved in grooming with the dominant aggressor and with the non-playback dominant following afternoons with simulated within-group conflict or matched control afternoons (N=8 groups).

*et al., 2008*; *Verbeek and de Waal, 1997*). Subordinate bystanders could be avoiding the aggressor to reduce the likelihood of redirected aggression, which parallels the main strategy employed in the immediate aftermath of contests by meerkat (*Suricata suricatta*) and rook (*Corvus frugilegus*) targets attempting to avoid renewed aggression (*Benkada et al., 2020*; *Kutsukake and Clutton-Brock, 2008*). Our results support previous research showing that within-group conflict can affect interactions beyond those between the protagonists and highlight that bystanders can employ different conflict-management strategies depending on the identity of the group members involved. Disagreements between two individuals can thus have wide-reaching and varied implications for affiliative behaviour,

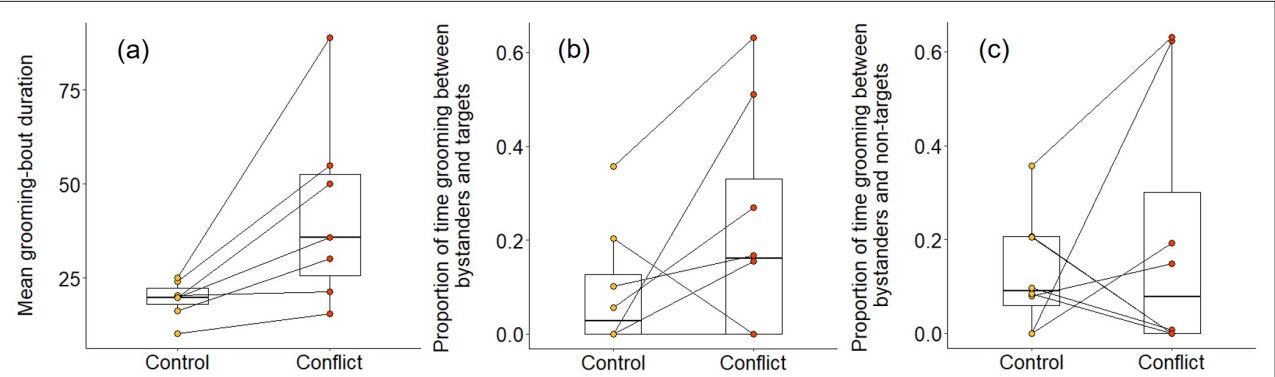

**Figure 6.** Delayed effect of experimentally increased within-group conflict on the grooming between subordinate bystanders. Compared to control afternoons, those with simulated additional foraging displacements between a dominant aggressor and a subordinate target resulted in (**a**) a greater mean duration of grooming bouts (s) between subordinate group members. There was (**b**) an indication that subordinate bystanders and simulated targets groomed for a greater proportion of time on conflict evenings compared to control ones, although the result was not statistically significant; (**c**) there was no equivalent treatment difference in the proportion of time that subordinates groomed with non-target subordinates. Shown in all panels are boxplots with the median and quartiles; whiskers represent data within quartiles ± 1.5 times the interquartile range. Values for each group are plotted separately (N = 8), with lines connecting data from the same group; in some instances, more than one group has the same value, hence the number of dashed lines can appear less than eight.

The online version of this article includes the following source data for figure 6:

**Source data 1.** Mean duration of grooming bouts between subordinate group members, and proportion of time grooming with the target and a non-target individual following afternoons with simulated within-group conflict or matched control afternoons (N=8 groups).

which underpins dyadic relationships and social structure in dwarf mongooses (*Kern and Radford, 2021*; *Kern and Radford, 2018*) and many other group-living species (*Cameron et al., 2009*; *Radford and Du Plessis, 2006*; *Silk et al., 2009*).

The changes in evening grooming patterns once the group had moved from their afternoon foraging areas demonstrate that individuals can retain information relating to earlier within-group conflict and use it when making later decisions about management-related behaviours. It is possible that hearing foraging displacements generates stress in bystanders, which could still be elevated on arrival at the sleeping burrow. But the specific reduction in grooming of perceived aggressors and not of non-playback dominants suggests decision-making about which groupmates to groom rather than a general byproduct of an altered physiological state. Numerous studies on a range of species have found changes in affiliative behaviour between various combinations of protagonists and bystanders in the minutes after within-group contests (*Aureli et al., 2002*; *de Waal, 2000*). It remains unknown whether, in those species, there might be delayed effects (as we have found) of those contests that do not result in immediate changes in affiliative behaviour. By using call playbacks, we did not alter the state of the individuals who were simulated to be the aggressor and target, and thus can rule out the possibility that differences in grooming result from experimentally induced satiation effects (which might have been the case if we had caused foraging displacements with the presentation of food items; *Sharpe et al., 2013*). Moreover, the use of playback simulations, rather than generation of actual contests, means that the delayed grooming effects are most likely driven by subordinate bystanders behaving differently towards perceived protagonists, rather than solicitation or rejection of grooming by the latter. Previous work has shown that dwarf mongooses can use vocal information to detect earlier cooperative contributions by groupmates and then reward them later (*Kern and Radford, 2018*); we now provide experimental evidence for delayed post-conflict management behaviour. It is thus increasingly apparent that mongooses, as well as many other social species, are constantly monitoring the behaviour and interactions of groupmates, and are using memories of what they learn to inform later decisions (*Tibbetts et al., 2020*; *Wittig et al., 2014*). The cognitive demands of tracking individuals and their behaviours, remembering that information, and using it when making decisions explain why social interactions within (*Dunbar and Shultz, 2007*) and between (*Ashton et al., 2020*) groups are believed to be strong drivers of animal intelligence.

Our experiments show that dwarf mongooses can extract information about within-group conflicts, and the identity of at least some protagonists, from vocal cues alone. This adds to a growing body of

work demonstrating the ability of social species to garner information acoustically about aggressive interactions (*Gouzoules et al., 1984*; *Slocombe et al., 2010*; *Slocombe and Zuberbühler, 2007*); for example, male little blue penguins had an increased heartrate after hearing vocalisations produced by winners of a contest compared to those produced by losers (*Mouterde et al., 2012*). Our findings also complement the small number of studies showing that social animals use vocalisations to assess the behaviour, such as the reliability (*Blumstein et al., 2004*) and cooperative contributions (*Kern and Radford, 2018*), of individually identifiable groupmates. Acoustic monitoring is beneficial as it allows information acquisition in environments where it would be difficult to do so visually (eg, in low-light and dense vegetation) or when group members are widely scattered and communication is needed over long distances (*Bradbury and Vehrencamp, 2011*). Moreover, acoustic information can be gathered at a relatively low cost: it can be done whilst still actively foraging (*Hollén et al., 2008*) and, in the case of aggressive encounters, at a safe distance that minimises the risk of the information-gatherer receiving any redirected aggression. Monitoring behaviours acoustically is likely not possible for all within-group interactions (eg, grooming) or in all social systems, but the calls commonly produced during and at the end of aggressive contests (*Bertram et al., 2010*; *Slocombe et al., 2010*) provide a valuable means for bystanders to inform subsequent decision-making.

We found clear evidence that subordinate bystanders engage in less grooming with simulated aggressors, but whether they increased their grooming with the simulated target is less clear-cut. There are several possible explanations for this difference in the strength of response exhibited to the two protagonists. First, all subordinates might be wary of the aggressor and so potentially reduce their grooming with that individual, whereas perhaps only those who are strongly bonded to the target might engage in extra grooming with it (*Fraser et al., 2009*; *Fraser et al., 2008*); any such target-related effect might be diluted by considering all subordinates in analyses. Strong within-group relationships are apparent in dwarf mongoose groups (*Kern and Radford, 2021*; *Kern and Radford, 2016*), but we do not have the power in this study to consider how relationship quality influences delayed post-conflict grooming. Alternatively, there could be selective attention towards high-ranking individuals (*Chance, 1967*): there may be higher selective pressure to discriminate vocalisations from dominant individuals cf those from other subordinates if the former are more important in terms of social relationships and status. In many primate species, for example, individuals focus attention on higher-ranking groupmates or those with whom they have an antagonistic relationship (*Keverne et al., 1978*; *McNelis and Boatright-Horowitz, 1998*), possibly to avoid aggression (*Schino and Sciarretta, 2016*). Since our simulated aggressors were dominants and our simulated targets were subordinates, the stronger effect of increased conflict on grooming with the former could reflect such an attention bias.

Another possible reason for the difference in grooming responses to aggressors and targets could be differences in the natural acoustic properties of aggressive growls and submissive squeals (*Gustison and Townsend, 2015*). In principle, squeals might encode less identity information than growls (*Owren and Rendall, 2003*; *Rendall et al., 1996*), although a number of studies have found that calls similar in structure and function to dwarf mongoose squeals are individually identifiable (*Cheney and Seyfarth, 1980*; *Fischer, 2004*; *Gouzoules et al., 1984*; *Slocombe and Zuberbühler, 2005*). In addition, our playback contained three growls and one squeal (to reflect natural foraging displacements), which could have made growls more salient or memorable and/or aided easier discrimination of the aggressor compared to the target. It might also be more cognitively demanding for receivers to discriminate the squeals from multiple subordinate individuals in a group, compared to growls, which are highly likely to come from one of the two dominant individuals. Finally, since contest-related vocalisations may vary depending on the severity of an attack (*Gouzoules et al., 1984*; *Slocombe and Zuberbühler, 2007*), it is possible that we used less salient squeals than growls in our playbacks (although both were recorded during natural foraging displacements). Future work is required to tease these possibilities apart.

In summary, our results demonstrate that dwarf mongooses can obtain information about within-group contests (including protagonist identity) acoustically, retain that information, and use it to inform decisions about conflict management with a temporal delay. Such a delay might be most apparent in situations where there is little opportunity for immediate post-contest affiliation (as is the case with foraging dwarf mongooses); it may also be most apparent when there is a cumulative build-up of unresolved conflict. Our results come from a single population of dwarf mongooses, but

we do not believe that there were any obvious biases introduced into the data collection that would limit the generalisability of the findings (see the end of 'Materials and methods' for an assessment of the STRANGEness of our test sample). There is increasing experimental evidence that social animals can remember past events and take these into account when deciding whether to get involved in a contest (*Borgeaud and Bshary, 2015*; *Cheney et al., 2010*; *Tibbetts et al., 2020*; *Wittig et al., 2014*); we demonstrate that this ability extends to post-conflict affiliative behaviour. More generally, our results showcase the importance of considering group-wide consequences of dyadic within-group interactions and of looking for effects beyond the immediate aftermath.

## Materials and methods

### Study site and population

We conducted our study on Sorabi Rock Lodge (24° 11'S, 30° 46'E), a private game reserve in the Limpopo Province, South Africa; full details are available in *Kern and Radford, 2013*. This is the site of the Dwarf Mongoose Research Project (DMRP), which has been studying a wild population of dwarf mongooses since 2011. At the time of the study (June to October 2019; non-breeding season), eight dwarf mongoose groups (mean ± SE group size: 12.3 ± 1.7, range: 5–16) were fully habituated to the close presence (<5 m) of human observers on foot. All the individuals in the population were identifiable, either through dye marks on their fur (blond hair dye applied using an elongated paintbrush) or natural features, such as scars. Individuals older than 1 year were classified as adults (*Kern et al., 2016*) data collection was focussed on adults as younger individuals are seldom involved in foraging displacements. Adults were sexed by observing ano-genital grooming (*Kern et al., 2016*) and classified as being either dominant (the male and female breeding pair) or subordinate; dominance status was established through observation of targeted aggression, scent marking, and reproductive behaviour (*Kern and Radford, 2013*; *Rasa, 1977*).

### Observational data collection

To determine the natural frequency of foraging displacements in our experimental period, we recorded all observer-detected occurrences of such behaviour during observation sessions; this included displacements that were seen and heard. The calculated rate is likely a conservative estimate as an observer could have missed a foraging displacement (particularly when the group was relatively widely scattered). We used data collected ad libitum as part of the long-term DMRP to assess the likelihood of particular dyads of individuals (aggressor–target: dominant–dominant, dominant–subordinate, subordinate–subordinate, subordinate–dominant) being involved in a foraging displacement.

To collect data on responses to natural foraging displacements, we conducted paired focal watches (conflict and control) of 2–3 min duration on 16 subordinate group members in six groups whilst they were foraging; conflict and control focal watches did not differ significantly in their duration (Wilcoxon signed-rank test: Z = 0.952, N = 16, p = 0.380). A conflict watch was carried out immediately after a foraging displacement was heard by the observer, whilst a control watch was carried out when there had been no foraging displacement (or any other agonistic interaction) for at least 10 min. We only carried out focal watches when the relevant mongoose was in a medium-cover habitat (20–60 % ground cover), weather conditions were calm (still or light breeze), there had been no alarm call (conspecific or heterospecific) in the previous 10 min, there had been no predator encounter or inter-group interaction for at least 30 min, and the focal individual was not on the periphery of the group. We abandoned focal watches, and repeated them later, if the focal individual stopped general foraging activities or if there was an alarm call within the first 2 min. Otherwise, we aimed to collect 3 min of uninterrupted data, but if a behavioural change or alarm call occurred between the second and third minute, then the focal watch was retained. Pairs of watches on the same focal individual were completed within 1 month (mean ± SE: 8.1 ± 2.7 days apart, range: 0–30 days); group composition always remained the same between a pair of watches, and a minimum of 1 hr was left between watches that were conducted on the same day. We watched nine individuals first in control conditions and seven first following a foraging displacement.

During each focal watch, we recorded behavioural data to a Dictaphone (ICD-PX312, Sony; Sony Europe Limited, Surrey, UK). Dwarf mongooses have two types of vigilance behaviour: vigilance scans, where individuals temporarily stop foraging in a head-down position to scan their surroundings (*Kern*

*et al., 2016*), and sentinel behaviour, where individuals cease foraging to scan from a raised position (minimum 10 cm above the ground level; *Kern and Radford, 2013*). Throughout each focal watch, we dictated the start and end point of each vigilance scan and sentinel bout, along with the occurrence of any grooming interaction with a groupmate. These data were used to calculate the proportion of time spent vigilant; no grooming occurred during these focal watches. No individual acted as a sentinel during the observational focal watches, and therefore the vigilance response measure was based on scan data only. We used a Wilcoxon signed-rank test to analyse the proportion of time vigilant in SPSS 24 (IBM Corp, 2016). Due to small sample sizes, we used the Monte Carlo repeated sampling method (based on 10,000 samples) to calculate an unbiased estimate of the exact p-value (*Mehta and Patel, 2011*).

## Experimental stimuli

We conducted two field-based repeated-measures experiments using playbacks to simulate the occurrence of conflict between group members. Each experiment involved the playback of 'conflict' and 'control' tracks. We recorded all calls for track creation when weather conditions were calm using a Marantz PMD660 professional solid-state recorder (Marantz America, Mahwah, NJ) connected to a handheld Sennheiser ME66 directional microphone (Sennheiser UK, High Wycombe, Buckinghamshire, UK; frequency response: 40–20,000 Hz) with a Rycote softie windshield (Rycote Microphone Windshields, Stroud, Gloucestershire, UK). The Marantz was set to record at 48 kHz with a 16-bit resolution, and files were saved in wav format. For conflict tracks, we recorded aggressive growls and submissive squeals opportunistically from natural foraging displacements or from conflicts induced by the presentation of a small amount of hard-boiled egg. Growls were recorded from either the dominant male or the dominant female in each group and squeals were recorded from a subordinate male or female in each group; all recorded calls came from foraging displacements where the dominant was the aggressor and the subordinate was the target. We recorded close calls, for use in both control and conflict tracks, from the same dominant and subordinate individuals whilst they were foraging. Recordings of all vocalisations were made 0.5–5 m from the relevant individual.

We formed 40 s playback tracks in Audacity (version 2.1.3) by extracting calls of good signal-to-noise ratios from original recordings and inserting them into ambient-sound recordings; ambient sound was recorded from the centre of the territory of the focal group on calm days and in the absence of dwarf mongooses. The first 36 s of each track (conflict and control) consisted of non-overlapping close calls from the relevant dominant and subordinate individual, with a rate of one close call every 6 s per individual. This rate of close calling falls within the natural range (*Kern and Radford, 2013*). For conflict tracks, the last 4 s consisted of a sequence of three growls from the dominant followed by one squeal from the subordinate; multiple growls and a single squeal reflect natural foraging displacements (personal observation). In control tracks, the last 4 s consisted of three close calls from the dominant followed by one close call from the subordinate, to match the number of vocalisations in conflict tracks. Individual tracks always contained vocalisations from same-sex individuals.

We created nine unique conflict and control tracks for each group. Given that the first 36 s of each track comprised close calls from the dominant and subordinate individual, we created three close-call sequences for each individual (each sequence contained six close calls), resulting in nine unique close-call combinations. For the conflict tracks, in which the last 4 s contained growls and a squeal, we created three different growl sequences for the dominant (each sequence consisted of three growls), which were each combined with three separate squeals from the subordinate. Lastly, for the final 4 s of the control tracks, we made three close-call sequences for the dominant (each sequence contained three close calls to match the number of growls in conflict tracks) and combined these with three different close calls from the subordinate. We applied a low-pass filter (set to 200 Hz) to all tracks to remove low-frequency disturbances.

We played back tracks from an iPhone (Apple, Cupertino, CA), connected to a Rokono B10 (London, UK) portable loudspeaker (frequency response: 90–20,000 Hz) concealed in vegetation. We set the amplitude to a sound-pressure level of 55 dB(A) at 1 m for close calls and growls, and 65 dB(A) at 1 m for squeals. This was the relevant amplitude of these vocalisations as determined by measurement of natural calls with a HandyMAN TEK 1345 sound-level meter (Metrel UK Ltd, Normanton, UK).

## Experiment 1 protocol

Experiment 1 was a complement to the observational focal watches (see 'Observational data collection'), aiming to test whether bystanders might garner information about within-group conflict solely from vocalisations and then adjust their immediate affiliative behaviour. We randomly selected 17 subordinate individuals (excluding those whose calls were used in the playback tracks) to receive the two treatments (conflict and control) on separate days and in a counterbalanced order. Each treatment was repeated two to three times per individual during the same observation session, using a different playback track each time, with a minimum of 10 min between repeats; for one individual, it was possible to run one of the treatments only once. We completed the two treatments for the same individual within 2 weeks of each other (mean ± SE: 2.8 ± 0.7 days apart, range: 1–11 days) and at the same time of day (either between 07:00 and 12:00 or between 12:30 and 17:30). The 17 focal individuals were from eight groups; for groups where there was more than one focal individual (N = 4 groups), we completed both treatments on one individual before moving on to the next.

We conducted playbacks when the focal individual was foraging in a medium habitat with little or no breeze and when the callers in the playback were not the focal individual's nearest neighbour (other pre-requisites are detailed in 'Observational data collection'). Where possible, we placed the loudspeaker in the general direction of the playback individuals. As soon as the playback finished, we conducted a 2–3 min focal watch; the mean duration of focal watches was not significantly different between treatments (Wilcoxon signed-rank test: Z = 1.397, N = 17, p = 0.168). Collection of vigilance and grooming data was identical to that for observational focal watches.

As for the natural foraging displacements (see 'Observational data collection'), we analysed the proportion of time spent vigilant; no grooming occurred in any focal watches. Since each treatment was repeated two to three times on an individual, we analysed the mean proportion of time spent vigilant with a Wilcoxon signed-ranks test. In 5 out of 94 trials, an individual acted as a sentinel. We therefore ran the vigilance response measures including and excluding this sentinel behaviour. The data reported in the 'Results' section are those excluding sentinel bouts, but qualitatively similar results were found for those including this behaviour.

## Experiment 2 protocol

Experiment 2 aimed to test whether there was a delayed effect of within-group conflict on affiliation between group members. We gave eight groups two treatments each on separate days, with treatment order counterbalanced between the groups. On conflict days, the perceived level of within-group conflict was increased during the afternoon by a playback of up to nine conflict tracks. On control days, perceived levels of within-group conflict were unmanipulated; up to nine control tracks were played back during the afternoon instead. There was no treatment difference in the number of natural foraging displacements that occurred throughout the afternoon (Wilcoxon signed-rank test: Z = 1.725, N = 8, p = 0.158). We completed the two treatments for the same group within 2 weeks of each other (mean ± SE: 3.3 ± 1.0 days apart, range: 1–9 days). Trials were only attempted when the weather conditions were suitable (not too windy or cold) and were abandoned if any major disturbances occurred during the afternoon (eg, predation attempts, inter-group interactions, multiple latrine events).

On a trial afternoon, we played back tracks from the centre of the foraging group approximately every 20 min during the 3 hr period before the group started moving to an evening sleeping refuge. There were five trials (two conflict, three control) where circumstances (eg, groups on the move, individuals foraging too far apart) prevented us from completing all nine planned playbacks in an afternoon (mean ± SE number of playbacks per trial: 8.5 ± 0.2, range: 6–9) before the group headed to their sleeping refuge. Once at the refuge (always termite mounds), we recorded all instances of adult grooming behaviour ad libitum until the mongooses went below the ground for the night; it is possible to collect data on all group members simultaneously because they are within a small area around the refuge compared to being scatted more widely when foraging (ie, in Experiment 1). Data collection involved dictating the identity of grooming partners and the start and end point of each bout. Periods of grooming data collection at the refuge (mean ± SE: 15.5 ± 2.3 min, range: 2–37 min) were not significantly different in duration between treatments (Wilcoxon signed-rank test: Z = 1.332, N = 8, p = 0.209).

To analyse the overall grooming data at the refuge (including grooming bouts >5 s; *Kern and Radford, 2018*), we constructed mixed models in RStudio 3.6.2 (R Core Team 2019) using the packages lme4 (*Bates et al., 2015*) and glmmTMB (*Brooks et al., 2017*). For all models, we included treatment as a fixed effect and nested Individual ID within Group ID as random effects to account for data from the same individuals and groups. Error distributions were chosen such that there were no deviations from normality or homoscedasticity, as checked by graphical examination of residual plots; certain response variables were transformed to meet the assumptions of parametric testing. To assess the significance of treatment (our one fixed effect), we compared a model containing treatment to a model without it (null model) using a likelihood ratio test (analysis of variance (ANOVA) model comparison, $\chi^2$ test). All tests were two-tailed and considered significant below an alpha level of 0.05.

We first ran a GLMM to assess whether there was a difference in the likelihood that adult individuals participated in grooming behaviour; our response measure was a binary term—whether the individual engage in any grooming (Yes or No) For those individuals that did participate in grooming, we ran additional models to understand this behaviour further. We first analysed in a GLMM the proportion of time that individuals spent grooming (summed grooming durations for each individual divided by the time available for grooming at the refuge, with the latter defined as the duration between the first and last grooming bout). We then considered whether the increase in proportion of time grooming was driven by a greater frequency (GLMM analysing the number of grooming interactions each individual was involved in, with log(duration) as an offset term to account for differences in the time available for grooming) or an increase in mean bout duration (LMM). We subsequently ran Wilcoxon signed-rank tests in SPSS 24 (as in 'Observational data collection' and 'Experiment 1 protocol') to consider the grooming behaviour between specific categories of group members (see 'Results').

## STRANGE framework

We have evaluated the STRANGEness of our test sample (*Webster and Rutz, 2020*) and believe that for the research topic in question there was minimal introduced bias. We worked with free-living animals from a wild population of dwarf mongooses, so no trapping or housing was involved in the study; all members of the study groups were habituated to close observer presence, and so no bias in random selection occurred due to variation in the ability to approach potential subjects. Focal individuals for observational data collection and Experiment 1 were randomly selected subordinate adults of both sexes from the study groups. Subordinates were chosen since the majority of foraging displacements occur between a dominant individual and a subordinate, and in Experiment 2, we were interested in comparing how bystanders groomed a perceived aggressor (one of the dominant pair) and the other dominant individual. For Experiment 2, we recorded all instances of adult grooming behaviour in the study groups. The population had not been exposed to these experiments previously.

## Acknowledgements

We thank B Rouwhorst and H Yeates for access to their land, I Carpenter, M Layton, J Linden, I Shan, and N Tegtman for their invaluable support and assistance in the field, M Aveling for the beautiful figure illustrations, and I Braga-Goncalves, S King, P Kennedy, M Mirville, and an anonymous referee for useful comments on the manuscript.

## Additional information

### Funding

| Funder | Grant reference number | Author |
| --- | --- | --- |
| H2020 European Research Council | 682253 | Andrew N Radford |

The funders had no role in study design, data collection and interpretation, or the decision to submit the work for publication.

## Author contributions
Amy Morris-Drake, Conceptualization, Data curation, Formal analysis, Investigation, Methodology, Project administration, Validation, Visualization, Writing - original draft, Writing – review and editing; Julie M Kern, Project administration, Resources, Supervision, Writing – review and editing; Andrew N Radford, Conceptualization, Funding acquisition, Methodology, Resources, Supervision, Writing – review and editing

## Author ORCIDs
Amy Morris-Drake ![ORCID] http://orcid.org/0000-0003-4243-4651
Andrew N Radford ![ORCID] http://orcid.org/0000-0001-5470-3463

## Ethics
The study was undertaken by permission from the Department of Environmental Affairs and Tourism, Limpopo Province (permit number: 001-CPM403-00013) and the Ethical Review Group, University of Bristol (University Investigator Number: UIN/17/074).

## Decision letter and Author response
Decision letter https://doi.org/10.7554/69196.sa1
Author response https://doi.org/10.7554/69196.sa2

---

# Additional files

## Supplementary files
• Transparent reporting form

• Source data 1. Source data for all statistical analyses (data for each separate analysis is provided in separate labelled worksheets).

## Data availability
Source data files have been uploaded for our figures and statistical tests.

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
