## [Decision Letter]

**Acceptance summary:**

This article will be of interest to behavioural ecologists studying aggression, within-group conflict, communication, and the use of social information. The study elegantly combines well-designed experiments with field observations to investigate the effects of within-group conflict on social behaviour. Specifically, it expands our understanding of social dynamics in group-living species by providing evidence that bystanders of within-group conflict may play a role in maintaining group cohesion. The findings provide a valuable contribution, and contrast, to existing work in this field.

**Decision letter after peer review:**

Thank you for submitting your article "Experimental evidence for delayed post-conflict management behaviour in wild dwarf mongooses" for consideration by *eLife*. Your article has been reviewed by two peer reviewers, and the evaluation has been overseen by Safi Darden as the Reviewing Editor and Christian Rutz as the Senior Editor. The following individual involved in the review of your submission has agreed to reveal their identity: Melanie Mirville (Reviewer #1).

The reviewers and editors have discussed their evaluations with one another, to reach a consensus recommendation. While we normally combine reviewer feedback into a consolidated set of essential revision requests, we have decided on this occasion to provide the reviewers' full reports below. In addition to addressing their detailed and constructive comments, please consider the following points when revising your article:

(1) We felt that an evaluation of the effects of increased vigilance on foraging success (satiation) would help support the conclusions drawn. If these additional analyses are impossible or impractical at this stage, we suggest acknowledging this and toning down language accordingly.

(2) We believe that terminology can be refined in places. For example, the temporal scale mentioned in the abstract and introduction is somewhat unclear – what exactly is the "delay" and what is its biological significance/justification? For example, what is known about physiological stress responses in this species (e.g., peak response and recovery time)? Furthermore, it may be more helpful to use the word "target", as opposed to "victim", as it is more neutral – we don't know anything about the psychology of the individuals involved in aggressive interactions.

(3) Please add more detail about the recording and playback equipment used (e.g., frequency response range) and provide information on whether there has been a comparison of stimulus spectrum and spectrum following playback (i.e., at 1 m), as it is important to demonstrate that these match, especially for the acoustic features thought to be biologically important.

(4) Please note that *eLife* has recently adopted the STRANGE framework, to help improve reporting standards and reproducibility in animal behaviour research. In your revision, please consider scope for sampling biases and potential limitations to the generalizability of your findings:

https://reviewer.elifesciences.org/author-guide/journal-policies

https://doi.org/10.1038/d41586-020-01751-5

*Reviewer #1 (Recommendations for the authors):*

Overall, the research is novel and presents unique results of interest to those studying social species. I believe that the manuscript would benefit greatly from including more information on the importance of interactions between individuals specific to mongooses to highlight the significance of changes in these behaviours, e.g. to set up why changes in grooming is important, what function does grooming have in mongooses specifically? why observe only bystanders in experiment 1, etc. I think the research is meaningful, but the overall cohesion of information presented in the introduction, results and discussion could be improved to increase the significance of this particular study.

Methods throughout – use consistent measurement terms, i.e. seconds, OR s, OR sec, same for minutes, hours, etc. Same with the use of numbers- spelled or use of numeric form.

Line 48 – is anxiety an anthropomorphised word? Is there another way of saying fear of conflict having occurred or for the potential to occur?

Line 51, 70, 89, 287 – perhaps specify post intragroup conflict throughout, as post-conflict could include intergroup which does have literature on longer-term outcomes on intragroup behaviour in social species.

Line 58 – suggest could be written more clearly, for example "were more likely to offer support in aggressive interactions to individuals they had groomed earlier, evidenced by differential movement towards grunt call playbacks during conflict."

Line 84 – what kind of valuable information i.e. specify the possible function of these calls?

Line 89 – suggest remove experimentally.

Line 103 – can you explain how vocalisations can act as reward to individuals?

Line 104 – can you explain the importance behind testing behaviour in only bystanders/subordinates, and not individuals involved in the conflict? Since this is your main hypothesis, I think more build up for the importance of this specific question is needed, i.e. highlight why we should know more about the effects of intragroup conflict on bystanders and their behaviour in the aftermath, what significance does this have?

Line 111, 119: prior to this, you mainly mention testing the delayed behavioural outcomes of intragroup conflict on individuals. I would suggest adding a results summary paragraph followed by subheadings with more specific results pertaining to each hypothesis, i.e. subheading 1 "experiment 1: "testing immediate behaviour outcomes of intragroup conflict on bystanders" lines 109-168 and subheading 2 "experiment 2: "testing delayed behaviour outcomes of intragroup conflict on bystanders" lines 168 onwards.

Line 125/420 – what is the function of a close call, can you explain how this is non-aggressive?

Line 160 – what is grooming at a distance? Did you measure this?

Line 195 – did you test grooming in those involved (i.e. the individuals used for the simulated vocalisations) vs bystanders?

Line 238, 243, 272 – change wording 'proportion of time' to 'time'.

Lines 241-248 – This can be written more succinct to make result clearer, suggest you combine the results for conflict and control evenings, i.e. 'less grooming among individuals on conflict evenings (evidenced by less time, state results, and fewer individuals, state results).

Line 253-261, 279-283 – move to discussion.

Line 295 – how did you decide that a conflict was not resolved?

Line 341 – new paragraph starting 'Alternatively…'

Line 349 – it may be cognitively demanding, but it also may not serve as much purpose to identify subordinate responses to aggressors. In other words, there may be a higher selective pressure to discriminate aggressive vocalisations due to the importance on individual relationships/social status, rather than subordinate responses.

Lines 359-364 – these statements are really important to highlight the significance of your findings, and I would suggest spending more time discussing these in your discussion to highlight the originality of your research.

Lines 365-368 – this statement about cognitive ability and intelligence comes somewhat out of the blue and may not be the best statement to end your article if you are intending to summarise the significance of this particular research, unless the link between post-conflict behaviour and social animal intelligence is discussed prior.

*Reviewer #2 (Recommendations for the authors):*

1. The modelling framework used to analyse the proportion of time grooming and the rate of grooming in linear models is not the most appropriate given the distribution of the data. Neither of these variables are normally distributed by definition (proportion time is bounded by 0 and 1, and a rate is a count of occurrences per unit time), and so log-transforming them is a bit like crowbarring them into a normal distribution. A more appropriate model to use for the proportion time data is one with a β error distribution which accommodates bounded variables (e.g. Smithson and Verkuilen 2006 https://doi.org/10.1037/1082-989X.11.1.54). Similarly, for the rate of grooming, a more appropriate model is one with a Poisson error structure that includes an offset (use 'offset' in R) of the exposure time (the time available to groom, log transformed) as an additional fixed effect. (See Crawley MJ. 2007. The R book, and numerous online forums).

2. Could means +/- SE for the treatment and control data be included in the results for Wilcoxon tests so that the reader can more easily evaluate the strength of the effect. Similarly, although you have provided output from linear models that includes effect estimates in a table, could you also include model estimates +/- SE in text with results, rather than just a chi-sq and p-value?

3. Can more information be provided on the Monte Carlo resampling method employed to generate P-values? This seems like an important part of analyses and subsequent interpretation of data (P-values are used to determine significance) and so more detail is needed on how this was performed.

4. It is not clear why you refer to using Akaike Information Criteria to evaluate fixed effects (L546) when there is no reporting of any AIC values for full and null models. Throughout the study, P-values are used to determine the significance of effects and so to also use AIC values conflates two very distinct statistical approaches. Can you remove reference to AIC values to avoid confusion?

5. L550 – you are testing a difference in the probability of grooming rather than 'number of adults that participated in grooming'?

6. It would help the reader to include short titles on panels of figures to quickly convey what the figure shows (or provide more informative y-axes titles). Currently some panels in the same plot are exactly the same in terms of axes (e.g. figure 5) and so it is not clear on first look what each panel shows and how it is different from others.

7. Was the focal individual in natural observations always unable to see the displacement (natural observations) or see the aggressor/victim in the playback (experiment 1)? Similarly, was the aggressor/victim in earshot when the playback of their calls was made (experiment 2)? Can you convince the reader a little more than your statement at L294-297 that hearing an interaction that does not match what an individual sees, or hearing yourself in an interaction you know you weren't involved in, does not change the behaviour of bystanders or simulated aggressors/victims.

8. The use of language like 'strong' and 'compelling' evidence (e.g. L287, L236) is a little overstated given the small sample sizes in the study; effects and p-values are likely to be very sensitive to sample size.

9. A little more information on why vigilance was measured as a behavioural response to within-group conflict would be welcome. The set up for measuring affiliative behaviour (grooming) was much clearer but why vigilance was used was less so. Does vigilance indicate being wary of conspecifics? How could you tease apart vigilance for predators versus that for conspecifics? Vigilance was also not measured as a delayed response as grooming was. Why was this? If because dwarf mongooses are very rarely vigilant for conspecifics at their sleeping refuge before they go down for the night, why was grooming measured during foraging in experiment 1 when it was noted that mongooses rarely groom in this context? Also, the increased vigilance result is comparatively neglected in the discussion of results compared to that of grooming, which feels unbalanced given that it is the primary result from the first set of natural observations and experiments.

---

## [Author Response]

The reviewers and editors have discussed their evaluations with one another, to reach a consensus recommendation. While we normally combine reviewer feedback into a consolidated set of essential revision requests, we have decided on this occasion to provide the reviewers' full reports below. In addition to addressing their detailed and constructive comments, please consider the following points when revising your article:(1) We felt that an evaluation of the effects of increased vigilance on foraging success (satiation) would help support the conclusions drawn. If these additional analyses are impossible or impractical at this stage, we suggest acknowledging this and toning down language accordingly.

The primary focus of the paper is post-conflict management behaviour (specifically, affiliative interactions) and, especially, whether that occurs with a delay since the conflict took place (lines 102–103). We collected data on vigilance behaviour immediately following natural and simulated foraging displacements simply as an indication that groupmates might be paying attention to vocal cues in these contexts; we have made that clearer (lines 127–128). As such, the vigilance result is not part of the main conclusions – we are not investigating how vocal cues of conflict affect vigilance and hence foraging – so we do not see a benefit in considering foraging success. Indeed, adding such analyses would over-emphasise this simple (background) element of the study. Perhaps we previously included too many vigilance results (i.e. an initial considering of proportion of time, with subsequent assessment of whether that difference was due to a difference in vigilance rate and/or bout duration), thus creating a false impression of the importance of that aspect. We have therefore reduced the vigilance part of the paper to just the overall proportion of time (lines 161–164), which is sufficient to make the point that bystanders are seemingly taking notice of the vocal cues of conflict (lines 164–166). Moreover, with respect to the natural foraging displacements and individually simulated conflicts, our Abstract had previously contained only mention of the vigilance results, and not the lack of any immediate post-conflict affiliative behaviour. This might also have wrongly over-emphasised the vigilance results, so we have now also included mention of the lack of evidence for immediate post-conflict affiliation in the Abstract (lines 18–19), before focusing on the delayed post-conflict management results (lines 19–25).

(2) We believe that terminology can be refined in places. For example, the temporal scale mentioned in the abstract and introduction is somewhat unclear – what exactly is the "delay" and what is its biological significance/justification? For example, what is known about physiological stress responses in this species (e.g., peak response and recovery time)? Furthermore, it may be more helpful to use the word "target", as opposed to "victim", as it is more neutral – we don't know anything about the psychology of the individuals involved in aggressive interactions.

We have made clearer what the delay is, in both the Abstract (lines 20–22) and the Introduction (lines 102–103). Not only is there a temporal gap of, on average, 37 min between the final simulated conflict and the grooming, but the whole group have moved from a foraging area to a sleeping burrow (lines 183–185). That means bystanders are not right next to protagonists as would be the case immediately following a conflict that they have visually observed. We do not know details of the physiological stress response in the study species so it is possible that, if hearing a conflict induced stress in a bystander then elevated stress might still be apparent at the later grooming time; we have included this possibility in the Discussion (lines 295–297). However, we find that, for example, bystanders decrease grooming of only the simulated dominant aggressor and not the other dominant in the group, which means the results are not simply the consequence of a general elevated stress level (lines 297–299). We have changed use of “victim” to “target” throughout the paper (e.g. lines 40, 148, 156 and 179).

(3) Please add more detail about the recording and playback equipment used (e.g., frequency response range) and provide information on whether there has been a comparison of stimulus spectrum and spectrum following playback (i.e., at 1 m), as it is important to demonstrate that these match, especially for the acoustic features thought to be biologically important.

We have added more detail about the recording and playback equipment, as requested (lines 456 and 493). We did not compare the stimulus spectrum and spectrum following playback and cannot get back to the field site in South Africa to do so at this stage.

(4) Please note that eLife has recently adopted the STRANGE framework, to help improve reporting standards and reproducibility in animal behaviour research. In your revision, please consider scope for sampling biases and potential limitations to the generalizability of your findings:https://reviewer.elifesciences.org/author-guide/journal-policieshttps://doi.org/10.1038/d41586-020-01751-5

We have added reference to the STRANGE framework in the Materials and methods (lines 583–595) and Discussion (lines 381–384), in line with *eLife*’s policies.

Reviewer #1 (Recommendations for the authors):Overall, the research is novel and presents unique results of interest to those studying social species. I believe that the manuscript would benefit greatly from including more information on the importance of interactions between individuals specific to mongooses to highlight the significance of changes in these behaviours, e.g. to set up why changes in grooming is important, what function does grooming have in mongooses specifically? why observe only bystanders in experiment 1, etc. I think the research is meaningful, but the overall cohesion of information presented in the introduction, results and discussion could be improved to increase the significance of this particular study.

We have included in the Introduction some background information on dwarf mongoose grooming interactions, which emphasises how fundamental they are to these societies (lines 108–112). We have added an explanation for the focus on bystanders (lines 121–123): since our experimental paradigm involves playback of protagonist calls (lines 140­–143), it is not ecologically relevant to consider how individuals would respond to playback of their own calls, as that is not a situation they would naturally encounter. We have restructured the start of the Discussion to emphasise the significance and novelty of our work: the first paragraph (line 269 ff.) showcases the importance of group-wide changes in affiliation as a consequence of within-group conflict between two individuals; the second paragraph (line 293 ff.) showcases the importance of remembering past events and using that information in later decision-making.

Methods throughout – use consistent measurement terms, i.e. seconds, OR s, OR sec, same for minutes, hours, etc. Same with the use of numbers – spelled or use of numeric form.

We have made sure we use consistent measurement terms with respect to seconds, minutes and hours. However, we do not believe that it is correct to use one of either spelled or numeric forms of numbers. Rather, in general, they should be spelled out in full for numbers up to nine, whereas they can be given as numerals for 10 and above, with the exceptions being (a) if a number is at the start of a sentence, when it should always be written in full, and (b) if a number is associated with a unit of measurement, when it can be a numeral regardless of the amount. We have stuck with these rules throughout.

Line 48 – is anxiety an anthropomorphised word? Is there another way of saying fear of conflict having occurred or for the potential to occur?

We have changed this to ‘stress’ here (line 56) and throughout, although previous reviewers of other papers have advised using ‘anxiety’ rather than ‘stress’ so there seems to be an ongoing debate about which is most appropriate.

Line 51, 70, 89, 287 – perhaps specify post intragroup conflict throughout, as post-conflict could include intergroup which does have literature on longer-term outcomes on intragroup behaviour in social species.

We have included ‘within-group’ for clarity in lines 58, 79 and 96–97. We have restructured the first Discussion paragraph and included ‘within-group’ in the relevant summary sentence there too (line 272).

Line 58 – suggest could be written more clearly, for example "were more likely to offer support in aggressive interactions to individuals they had groomed earlier, evidenced by differential movement towards grunt call playbacks during conflict."

We have changed the wording as suggested to improve clarity (lines 64–67).

Line 84 – what kind of valuable information i.e. specify the possible function of these calls?

We have included that the valuable information might be that within-group conflict has occurred, which group members were involved in the contest and who might have won (lines 95–97).

Line 89 – suggest remove experimentally.

We have removed ‘experimentally’ (lines 102–103).

Line 103 – can you explain how vocalisations can act as reward to individuals?

Apologies for the confusion. The vocalisations don’t act as a reward – rather, groupmates detect contributions to cooperative sentinel behaviour through the vocalisations of those individuals engaged in that activity, and then reward them later. We have rephrased for greater clarity but moved this sentence to the Discussion due to another comment (lines 311–313).

Line 104 – can you explain the importance behind testing behaviour in only bystanders/subordinates, and not individuals involved in the conflict? Since this is your main hypothesis, I think more build up for the importance of this specific question is needed, i.e. highlight why we should know more about the effects of intragroup conflict on bystanders and their behaviour in the aftermath, what significance does this have?

As we highlight in an Introduction paragraph, initial work on post-conflict behaviour focused on protagonists (lines 39–44) before, more recently, turning also to bystanders (lines 44–45); we have provided additional explanation about the importance of considering bystanders (lines 45–49). Because our experimental paradigm involves playback of calls of protagonists (lines 140–143, 178–180), it is not ecologically relevant to consider how those individuals would respond to playback of their own calls, as that is not a situation they would naturally encounter. Hence, we have focused on the responses of bystanders and made this logic clearer (lines 121–123). We would argue that our main hypothesis is not so much the response of bystanders per se, but the possibility of a delayed response to within-group conflict (lines 102–103).

Line 111, 119: prior to this, you mainly mention testing the delayed behavioural outcomes of intragroup conflict on individuals. I would suggest adding a results summary paragraph followed by subheadings with more specific results pertaining to each hypothesis, i.e. subheading 1 "experiment 1: "testing immediate behaviour outcomes of intragroup conflict on bystanders" lines 109-168 and subheading 2 "experiment 2: "testing delayed behaviour outcomes of intragroup conflict on bystanders" lines 168 onwards.

We have included mention of immediate, as well as delayed responses, in the penultimate sentence of the Introduction (lines 119–121) so that both elements of the study are set-up just before the Results. [We have moved reference to our previous work on delayed rewarding of cooperative behaviour to the Discussion (lines 310–312), so there is less of a focus on just the delayed element of our current work.] We have included subheadings in the Results as suggested (lines 126 and 174).

Line 125/420 – what is the function of a close call, can you explain how this is non-aggressive?

We had previously included additional information about close calls in the Methods, but have moved this up to first mention for greater clarity (lines 144–146). Close calls are low-amplitude vocalisations given continuously whilst foraging and likely serve to enable groupmates to stay in contact with one another; there is no evidence that they have an aggressive function.

Line 160 – what is grooming at a distance? Did you measure this?

Since this was somewhat of a random idea (there is no previous evidence of ‘grooming-at-a-distance’ in mongooses), it was only a single line in the Results, there was no significant effect and it was not something we measured for Experiment 2, we have decided to remove this from the manuscript. It does not change anything about our conclusions or our Discussion points.

Line 195 – did you test grooming in those involved (i.e. the individuals used for the simulated vocalisations) vs bystanders?

Our experimental paradigm entailed playback of calls to simulate a conflict, which has the advantages of not altering the state of the protagonists and of meaning that the change in behaviour by bystanders is unlikely to result from solicitation/rejection of grooming by simulated aggressors or targets (lines 303–310). Using playbacks, it would not be valid to consider the behaviour of the simulated protagonists because responding to hearing their own calls is not an ecologically valid situation. It is why we focused our data collection on bystanders (see earlier response).

Line 238, 243, 272 – change wording 'proportion of time' to 'time'

Changing the wording as suggested subtly changes the meaning: ‘time’ could be read as an absolute amount whereas we assessed the proportion of available time available for grooming that was spent engaged in that activity. So, we have left the wording as is.

Lines 241-248 – This can be written more succinct to make result clearer, suggest you combine the results for conflict and control evenings, i.e. 'less grooming among individuals on conflict evenings (evidenced by less time, state results, and fewer individuals, state results).

We have written this more succinctly as suggested (lines 238–243).

Line 253-261, 279-283 – move to discussion

We have moved these elements to the first paragraph of the Discussion (lines 272–284).

Line 295 – how did you decide that a conflict was not resolved?

Our choice of wording here was not ideal: we cannot know about conflict resolution from our data. We have therefore rephrased as: ‘… of those contests that do not result in immediate changes in affiliative behaviour.’ (line 302–303).

Line 341 – new paragraph starting 'Alternatively…'

We have started a new paragraph here as suggested, but rephrased what is now the opening sentence for clarity (line 360 ff.).

Line 349 – it may be cognitively demanding, but it also may not serve as much purpose to identify subordinate responses to aggressors. In other words, there may be a higher selective pressure to discriminate aggressive vocalisations due to the importance on individual relationships/social status, rather than subordinate responses.

We have included this possibility in the Discussion paragraph (lines 351–353).

Lines 359-364 – these statements are really important to highlight the significance of your findings, and I would suggest spending more time discussing these in your discussion to highlight the originality of your research.

We have altered the structure of the start of the Discussion to help highlight the significance of our results. The first paragraph now focuses on the demonstrated changes in affiliation involving bystanders (line 269 ff.), emphasising the importance of both the group-wide consequences of within-group conflict between two individuals and changes in grooming, which underpins social relationships and structure in many species (lines 285–291). The second paragraph now focuses on the delayed nature of the affiliative changes (line 293 ff.), emphasising the importance of social monitoring and an ability to remember past events when making decisions (lines 310–319).

Lines 365-368 – this statement about cognitive ability and intelligence comes somewhat out of the blue and may not be the best statement to end your article if you are intending to summarise the significance of this particular research, unless the link between post-conflict behaviour and social animal intelligence is discussed prior.

We agree that this was not the best way to end our paper, but we do think it an important point to make. So, we have moved this statement to the end of the second paragraph of the Discussion (lines 316–319) – which now focuses on the delayed aspect of the behavioural changes – where it helps to emphasise the general significance of our results. We now finish the Discussion with a more general point that relates directly to our findings (lines 387–389).

Reviewer #2 (Recommendations for the authors):1. The modelling framework used to analyse the proportion of time grooming and the rate of grooming in linear models is not the most appropriate given the distribution of the data. Neither of these variables are normally distributed by definition (proportion time is bounded by 0 and 1, and a rate is a count of occurrences per unit time), and so log-transforming them is a bit like crowbarring them into a normal distribution. A more appropriate model to use for the proportion time data is one with a β error distribution which accommodates bounded variables (e.g. Smithson and Verkuilen 2006 https://doi.org/10.1037/1082-989X.11.1.54). Similarly, for the rate of grooming, a more appropriate model is one with a Poisson error structure that includes an offset (use 'offset' in R) of the exposure time (the time available to groom, log transformed) as an additional fixed effect. (See Crawley MJ. 2007. The R book, and numerous online forums).

We have reanalysed the two mentioned datasets using the suggested approaches. We have therefore adjusted the relevant parts of the Methods (lines 571–578) and Results (lines 205–208), as well as Table 1 (lines 223–228). The results (and conclusions) remain qualitatively the same as in the original submission.

2. Could means +/- SE for the treatment and control data be included in the results for Wilcoxon tests so that the reader can more easily evaluate the strength of the effect. Similarly, although you have provided output from linear models that includes effect estimates in a table, could you also include model estimates +/- SE in text with results, rather than just a chi-sq and p-value?

We are wary of providing means +/- SE values in association with non-parametric tests as there are many readers who will view that as a disconnect; it is why we plot medians and IQRs in our figures. Including additional statistical information in parentheses makes text sentences increasingly difficult to read. Given that the associated table with the full information is part of the main manuscript (rather than being in Supplementary Information), our preference is to leave the reporting as is (this also minimises repetition between main text and table).

3. Can more information be provided on the Monte Carlo resampling method employed to generate P-values? This seems like an important part of analyses and subsequent interpretation of data (P-values are used to determine significance) and so more detail is needed on how this was performed.

We have provided more information on this resampling method (lines 446–448).

4. It is not clear why you refer to using Akaike Information Criteria to evaluate fixed effects (L546) when there is no reporting of any AIC values for full and null models. Throughout the study, P-values are used to determine the significance of effects and so to also use AIC values conflates two very distinct statistical approaches. Can you remove reference to AIC values to avoid confusion?

We used AIC simply to check that removal of treatment (our one fixed effect) didn’t improve the fit of the model (which it never did in relevant cases). We agree that mention of it is therefore confusing and so have removed this from the paper.

5. L550 – you are testing a difference in the probability of grooming rather than 'number of adults that participated in grooming'?

Yes, that is correct. We have reworded for clarity (lines 569–570).

6. It would help the reader to include short titles on panels of figures to quickly convey what the figure shows (or provide more informative y-axes titles). Currently some panels in the same plot are exactly the same in terms of axes (e.g. figure 5) and so it is not clear on first look what each panel shows and how it is different from others.

We have provided more informative y axis labels for Figures 4–6 so there are no longer any panels with the same labels.

7. Was the focal individual in natural observations always unable to see the displacement (natural observations) or see the aggressor/victim in the playback (experiment 1)? Similarly, was the aggressor/victim in earshot when the playback of their calls was made (experiment 2)? Can you convince the reader a little more than your statement at L294-297 that hearing an interaction that does not match what an individual sees, or hearing yourself in an interaction you know you weren't involved in, does not change the behaviour of bystanders or simulated aggressors/victims.

It is impossible for a single observer to track the exact whereabouts, behaviour and likely lines of sight or auditory input of all group members simultaneously, as the mongooses are somewhat scattered when foraging. We would therefore be wary about making additional claims along those lines. Our point was that, compared to supplementary feeding (another potential method of inducing/simulating foraging displacements), using playbacks was much less likely to affect the state of the individuals perceived by others to be involved. We have reworded the relevant sentence to focus on just state (lines 303–307).

8. The use of language like 'strong' and 'compelling' evidence (e.g. L287, L236) is a little overstated given the small sample sizes in the study; effects and p-values are likely to be very sensitive to sample size.

We have removed use of these terms (line 233).

9. A little more information on why vigilance was measured as a behavioural response to within-group conflict would be welcome. The set up for measuring affiliative behaviour (grooming) was much clearer but why vigilance was used was less so. Does vigilance indicate being wary of conspecifics? How could you tease apart vigilance for predators versus that for conspecifics? Vigilance was also not measured as a delayed response as grooming was. Why was this? If because dwarf mongooses are very rarely vigilant for conspecifics at their sleeping refuge before they go down for the night, why was grooming measured during foraging in experiment 1 when it was noted that mongooses rarely groom in this context? Also, the increased vigilance result is comparatively neglected in the discussion of results compared to that of grooming, which feels unbalanced given that it is the primary result from the first set of natural observations and experiments.

As explained in response to General Comment #1, vigilance was measured purely as an indicator of whether the dwarf mongooses appear to be paying attention to vocal cues of within-group conflict. We have reduced the number of results presented with respect to vigilance, to reduce the emphasis. We did not measure vigilance in Experiment 2 because we had already established from Experiment 1 that the mongooses pay attention to the relevant playbacks. Given our vigilance measure is effectively just a check for attention, we do not believe it warrants detailed discussion; it is not the focus of the study.